# How Functional Lipids Affect the Structure and Gating of Mechanosensitive MscS-like Channels

**DOI:** 10.3390/ijms232315071

**Published:** 2022-12-01

**Authors:** Vanessa Judith Flegler, Tim Rasmussen, Bettina Böttcher

**Affiliations:** Rudolf Virchow Center and Biocenter, University Würzburg, 97080 Würzburg, Germany

**Keywords:** mechanosensitive, MscS-like channels, functional lipids, lipid-protein interactions

## Abstract

The ability to cope with and adapt to changes in the environment is essential for all organisms. Osmotic pressure is a universal threat when environmental changes result in an imbalance of osmolytes inside and outside the cell which causes a deviation from the normal turgor. Cells have developed a potent system to deal with this stress in the form of mechanosensitive ion channels. Channel opening releases solutes from the cell and relieves the stress immediately. In bacteria, these channels directly sense the increased membrane tension caused by the enhanced turgor levels upon hypoosmotic shock. The mechanosensitive channel of small conductance, MscS, from *Escherichia coli* is one of the most extensively studied examples of mechanically stimulated channels. Different conformational states of this channel were obtained in various detergents and membrane mimetics, highlighting an intimate connection between the channel and its lipidic environment. Associated lipids occupy distinct locations and determine the conformational states of MscS. Not all these features are preserved in the larger MscS-like homologues. Recent structures of homologues from bacteria and plants identify common features and differences. This review discusses the current structural and functional models for MscS opening, as well as the influence of certain membrane characteristics on gating.

## 1. Introduction

Mechanosensitive (MS) channels open in response to changes in the membrane caused by mechanical stimuli. In higher organisms, diverse superfamilies of MS channels are involved in, e.g., in the sensations of touch and hearing, blood pressure regulation, or the perception of gravity [1]. Prominent representatives of MS channel families include the eukaryotic PIEZO channels that are involved in touch sensation [2], OSCA/TMEM63 channels that regulate Ca^2+^ uptake upon osmotic stress in plants and mammals [3], and certain members of the two-pore potassium channel family which finetune electric properties of the membrane [4,5]. However, already in 1981, advances of the patch clamp technique paved the way for the discovery and investigation of two distinct bacterial MS channels [6,7,8]: The mechanosensitive channel of large conductance (MscL) and the mechanosensitive channel of small conductance (MscS) [9,10]. While these two channels fulfil a similar function, they are structurally diverse and exhibit different gating characteristics. Furthermore, they differ in their distribution among species: MscL and homologues are restricted to bacteria and fungi. In contrast, MscS and homologues are, in addition to bacteria and also found in giga viruses, archaea and in eukaryotic organisms like protozoa, fungi and plants [11,12,13].

In bacteria, MscL and MscS play a major role in the ability of bacteria to adapt quickly to osmotic changes in their surroundings. Osmotic forces can have detrimental effects on the viability of the cells: A rapid drop in the osmolarity of the surrounding environment leads to an influx of water into the cell (Figure 1). Upon such a hypoosmotic shock, MS channels sense elevated levels of membrane tension which are caused by the increased pressure within the cell, gate, and release osmolytes. Thereby, they prevent cell lysis, and have earned the nickname “safety valves” [11]. MscL and MscS have both been extensively biophysically studied, and thus contributed substantially to our understanding of mechanosensation. They serve as valuable examples to reveal the interplay between channels with their lipidic environment [14,15].

In the last few years, the role of bacterial MscS-like channels for infection was more closely examined, pushing these channels towards a therapeutical interest. Their involvement in bacterial pathogenesis and antibiotic susceptibility was recently reviewed in much detail by Sidarta et al. [16] Specifically, two MscS homologues from *Campylobacter jejuni*, known to be a major source for gastroenteritis in humans, were shown to be required for protection during osmotic shifts [17]. Additionally, *Francisella tularensis*, which causes waterborne tularemia outbreaks, was shown to survive the osmotic change that happens during the transition from the host to freshwater through a MscS homologue [18]

Application of electron cryomicroscopy (cryo-EM) revealed intricate ligand networks in MscS [19,20,21] and showed the effects of lipid depletion on the conformation [21,22]. In this review, we focus on the recent findings regarding the role of lipids for the gating of MscS and discuss the proposed gating mechanisms. Furthermore, we include the latest studies on the larger MscS paralogue YnaI [23,24,25] and MscS-like plant channels MSL1 and Flycatcher1 [26,27,28] in our evaluations, highlighting both conserved features as well as dissimilarities between different MscS homologues.

## 2. MscS and MscS-like Channels

### 2.1. Architecture of MscS

MscS is the smallest representative of the MscS-like channel family in *Escherichia coli*. It comprises a homoheptamer with 286 amino acids per subunit. The complex can be divided into a large cytosolic domain, forming the vestibule, and a transmembrane (TM) domain (TMD) [29,30] (Figure 2A and Figure 3). While the cytosolic domain provides lateral portals for the channel and also determines the slight anion preference of MscS [9,31,32,33], the TMD is responsible for tension sensing and gating. MscS has three TM helices per subunit, of which the outer helices TM1 and TM2 constitute the so-called ‘sensor paddle’ helix bundle. This paddle is tilted away from the pore axis, which results in an unusually tapered shape. The helix TM3 is divided by a kink into the pore lining helix TM3a and the domain connecting helix TM3b. The latter lies almost parallel to the membrane plane and connects to the cytosolic domain, which comprises a β-domain, an αβ-domain, and a β-barrel at the C-terminus that provides intra-subunit stabilization and oligomerization [34]. The organization of the TM helices leads to pockets between the pore and paddle helices of adjacent subunits [35] (Figure 2A). No crystal structure of MscS resolved the 20 N-terminal residues on the periplasmic side of the membrane [29,30,32,36]. Furthermore, it was initially presumed that the TMD is entirely embedded in the membrane bilayer [37,38]. Later, cryo-EM structures of MscS in membrane mimetic systems showed that only the periplasmic half of the TMD is embedded in the membrane and that the larger part of the pore protrudes from the membrane on the cytosolic side [20,22,39]. So far, the architecture of MscS represents the most basic shared structural unit for members of the MscS-like channel family. Furthermore, structures of MscS homologues from *Helicobacter pylori* [36] and *Thermoanaerobacter tengcongensis* [32] were obtained that show the overall architecture of *E. coli* MscS.

A first crystal structure of open MscS was obtained by introducing the point mutation A106V (location shown in Figure 2C). The opening of MscS causes no notable changes in the vestibule, but the TM helices rearrange significantly [40]: The sensor paddle helices TM1 and TM2 rotate as a rigid body and increase their tilt with respect to the membrane plane (Figure 3). The pore helix TM3a rotates around its own helical axis, pivots around the kink connecting to the helix TM3b, and shifts outwards. The paddle reorganizations drag the pore more into the plane of the membrane, and the TMD appears flatter than in the closed conformation. Noteworthy, the membrane cross section of MscS enlarges only on the periplasmic side as the N-terminal domains move outwards whereas the diameter at the cytosolic side stays constant. Open conformations of MscS were also obtained under delipidating conditions [21,35,36], which all resemble that of the first open crystal structure using the A106V mutant [40]. The conformational state of MscS is characterized by the alignment of the sensor paddle with the helix TM3b: An open pore is observed when the cytosolic (N-terminal) end of the paddle helix TM2 interacts with the C-terminal end of helix TM3b of the same subunit, while a closed pore is linked to the interaction of the helices TM2 with helix TM3b of the adjacent subunit (Figure 3) [22]. 

Apart from its closed and open conformation, MscS enters two further non-conductive states under a prolonged pressure stimulus–a phenomenon called adaptation [41,42]. While one population, termed “desensitized”, can be activated again by application of a stronger pressure stimulus, the other population requires a resting time of several minutes in the absence of pressure to be activated again (“inactivated”) [43]. Although this behavior was known for a long time, an experimental structure of the inactivated conformation was only obtained recently [22]. In this study, persisting membrane tension was imitated by removing lipids from MscS containing nanodiscs with β-cyclodextrin (βCD): Nanodiscs are nanoparticles comprised of a lipid bilayer surrounded by α-helical membrane-scaffold proteins (MSPs). Depending on the kind of MSPs that are used for nanodisc formation, different sizes can be obtained [44,45,46,47]. The authors rationalized that lipid removal from the nanodiscs would result in tension, because the diameter of the nanodisc should be determined by the membrane scaffold proteins, and therefore, the remaining lipids would need to stretch to still fill the predetermined area of the nanodisc. The pore diameter in this inactivated state is the same as for the closed conformation, concomitant with the observation that helix TM2 aligns with helix TM3b of the adjoining subunit as mentioned above, but the paddle helices are tilted by 17° more, resulting in an even more flattened TMD than observed for the open conformation (Figure 3).

### 2.2. Structure of YnaI and Further Larger MscS-like Homologs

In addition to MscS, five further MscS-like channels are present in *E. coli*, suggesting they offer the cell the possibility for a tailored response to the hypoosmotic shock [42,48,49]. These MscS-like channels–YnaI, YbdG, YbiO, YjeP, and MscK–have all been shown to be mechanosensitive, differing in their gating characteristics and sizes mainly with respect to the number of TM helices [50]. However, they play only a minor role in osmoprotection, suggesting additional functions [51,52,53,54,55]. The medium-sized channel YnaI has two additional TM helices compared to MscS, which form a second helix bundle (Figure 2D and Figure 4, Table 1). They are similarly tilted as in MscS but are shifted against the inner helix bundle comprised of helices TM1 and TM2, which creates a large indentation on the periplasmic side [23,24]. 

All (putatively) closed YnaI structures so far were obtained in different reconstitution systems. The complex was prepared either in the presence of classic detergents–solubilized with n-dodecyl-β-d-maltoside (DDM) and reconstituted into Amphipols A8-35 [56], or with detergent lauryl maltose neopentyl glycol (LMNG) (Figure 2E, left) [24]–or in the absence of classic detergents using the membrane-active polymers diisobutylene/maleic acid (DIBMA) (Figure 2E, middle) [23] or styrene/maleic acid (SMA) 2000 (Figure 2E, right) [25]. While the cytosolic domains are all essentially identical, there are substantial variations seen within the TMD. The outermost N-terminal helix TM-2 (“minus 2”, corresponding to the nomenclature of MscS, Figure 2A) is best resolved in the sample solubilized in LMNG, while this helix as well as the loop connecting to helix TM-1 (“minus 1”) in YnaI, prepared with DIMBA copolymer shows only density fragments and is therefore probably incorrectly assigned. YnaI in Amphipols A8-35 and in SMA2000 only resolve the innermost helices TM2 and TM3 of the TMD. However, even the paddle helix TM2 shows significant differences: its orientation is largely identical to the structures of YnaI obtained in Amphipols, DIBMA and LMNG but tilted by ~15° in the structure of YnaI in SMA2000 (Figure 2E, right). This was linked to the position of the phosphate head group of a lipid bound in the pockets of YnaI [25]. Interestingly, all YnaI structures show a coordinated lipid that is differently orientated and positioned (Figure 2E) which underpins the sensitivity of the conformational state towards the preparation technique. 

MSL1 and Flycatcher1 are examples of larger MscS homologues found in plants, and their structures have recently been determined [26,27,28] (Table 1, Figure 4). MSL1 from *A. thaliana* shows a similar organization as YnaI regarding the five TM helices per subunit and the cytosolic domain except for a missing β-barrel at the C-terminus. Flycatcher1 is involved in touch sensation in *Dionaea muscipula* (Venus flytrap), but homologs are also expressed in mechanosensory cells of other carnivorous plants [57,58]. While the structure of the cytosolic vestibule bears a strong resemblance to that of *E. coli* MscS and YnaI, and *A. thaliana* MSL1, the TMD shows two major differences: Firstly, it has six TM helices per subunit, which locates the N-terminus on the cytosolic side of the membrane. Secondly, the TM helices 1 and 2 (corresponding to the nomenclature of MscS, Figure 2A) extend on the cytosolic side as flexible linkers, which can adopt two distinct conformations that are related to the functional states of the channel [28]. The cytosolic ends of the helix TM2 aligns with helix TM3b of the same subunit and thus, it should qualitatively represent the open conformation of the channel but based on MD simulations the authors concluded that the obtained structure probably does not represent the fully open state. An open form of *A. thaliana* MSL1 was obtained with the mutation A320V [27], which resembles the equivalent mutation that produced the first open MscS structure [40]. However, in the open conformation of MSL1, the kink that separates the helices TM3a and TM3b in the closed conformations of MscS-like channels, vanishes, resulting in one straight helix. This is remarkably different to MscS, where the kink separating the pore helices TM3a and TM3b is maintained in the open conformation [21,35,36,40]. It should be mentioned that straightening of the helix TM3 in the open state was also predicted for MscS based on early MD simulations [43].

An attempt to obtain an open conformation of *E. coli* YnaI made use of lysophosphatidylcholine (LPC) [23]. LPC was shown to activate the bacterial MS channels MscL and MscS [59,60,61,62], and also non-bacterial MS channels like TRAAK and TREK-1 [63]. Doping proteoliposomes harboring YnaI with LPC before extraction of the channels with DIBMA copolymer resulted in a structure of YnaI with an altered TMD [23]: The periplasmic indentation is significantly decreased and the TMD appears overall flatter. The pore helices TM3a bend away from the pore axis in the region of a glycine-rich GGxGG motif that is unique to YnaI among the bacterial MscS-like channels and this finding was supported by the characterization of pore mutations that either stabilize the closed or the open conformation [23]. This novel form was termed open-like and probably represents a sub-conducting form, because computational electrophysiology showed a reduced conductance compared to the value determined experimentally [23,53]. The fully open YnaI structure may be similar to the open form of *A. thaliana* MSL1 A320V [27], because of the GGxGG motif in the pore helix found in both channels and the conserved overall architecture of their closed forms [26,27]. However, it is also conceivable that the absence of the β-barrel in *A. thaliana* MSL1 is linked to the loss of the kink between the pore helices TM3a and TM3b.

**Table 1 ijms-23-15071-t001:** Selected properties of MscS and larger homologues with existing models. Not all available models are listed. #—number; SU—subunit; n.r.—not reported; n.m.—not modelled.

Channel	Organism	# TM Helices/SU	# Amino Acids/SU	Conductance (nS)	PDB	EMDB	Reconstitution	Putative State	Observed Lipids	Mutant?	Struct. Ref.
MscS	*Helicobacter pylori*	3	274	n.r.	4HW9	-	DDM	Closed	-	-	[33]
MscS	*Thermoanaerobacter tencongensis*	3	282	0.05–0.13 [29,52]	3T9N	-	Triton-X100	Closed	-	-	[29]
MscS	*Escherichia coli*	3	286	1.25 [28,39]	2OAU	-	Fos-choline 14	Closed	-	-	[26,27]
					2VV5	-	Fos-choline 14	Open	-	A106V	[37]
					4AGF	-	DDM	Open	-	L124C	[53]
					4HWA	-	DDM	Open	-	-	[33]
					5AJI	-	DDM	Open	Aliphatic chains	D67C	[32]
					6RLD	4919	MSP1E3D1 + Azolecin	Closed	3 per SU + aliphatic chains	-	[16]
					6PWN	20508	MSP1E3D1 + PC:PG (4:1)	Closed	2 per SU + aliphatic chains	-	[17]
					6UZH	20959	Peptidiscs	Closed	-	-	[54]
					6VYK	2146	MSP1E3D1 + PC-18:1	Closed	1 per SU + aliphatic chains	-	[19]
					6VYL	21463	MSP1E3D1 + PC-10	Subconduct.	-	-	[19]
					6VYM	21464	MSP1E3D1 + PC-18:1 + βCD	Adapted (inactivated)	-	-	[19]
					7OO0	13003	DDM	Open	1 per SU + DDM	-	[18]
					7OO6	13006	DDM + Azolectin	Closed	1 per SU + DDM	-	[18]
					7ONJ	12996	LMNG	Open	1 per SU + DDM + LMNG	-	[20]
YnaI	*Escherichia coli*	5	343	0.1 [53]	5Y4O	6805	Amphipols A8-35	Closed	-		[56]
					6ZYD	11557	DIBMA	Closed	1 per SU	-	[23]
					6ZYE	11560	DIBMA + LPC	Subcond. /unknown	-	-	[23]
					6URT	20862	LMNG	Closed	1 per SU	-	[24]
					7N4T	24177	SMA2000	Closed	1 per SU	-	[25]
MSL1	*Arabidopsis thaliana*	5	497	1.2 [64]	6LYP	30017	digitonin	Closed	-	-	[26]
					6VXM	21444	glyco-diosgenin	Closed	n.m.	-	[27]
					6VXN	21445	glyco-diosgenin	Open	n.m.	A320V	[27]
					6VXP	21447	MSP2NS + soybean polar lipid extract	Closed	n.m.	-	[27]
FLYC1	*Dionaea muscipula*	6	753	0.16–0.27 [28,57]	7N5D	24186	glyco-diosgenin	(not fully) open	n.m.	-	[28]
					7N5F	24188	glyco-diosgenin	“down”	-	-	[28]
					7N5G	24189	glyco-diosgenin	“up”	-	-	[28]

## 3. The Force-From-Lipids Principle

MS channels do not sense the increased turgor that is caused by a hypoosmotic shock, but the membrane tension resulting from it [65,66]. Furthermore, no additional cofactors or proteins are involved in sensing, because MS channels, solubilized, purified, and reconstituted into artificial membranes, retain their mechanosensitivity [10,31]. Therefore, the force-from-lipids principle was early advocated, stating that anisotropic forces of the membrane and their changes provide the gating force [67]. Anisotropic forces across the lipid bilayer are a consequence of the self-assembly of lipid molecules into a bilayer: While there is attraction between the glycerol groups of two adjacent lipid molecules, the fatty acid chains and the phosphate moieties of two lipid molecules repel each other. This leads to an anisotropic transbilayer pressure profile, and tension is highest in the region of the glycerol groups [68,69,70,71] (Figure 5A). Alteration of this pressure profile by stretching the membrane can provide the energy for a structural reorganization of an embedded protein [72]. The transbilayer pressure profile has usually been determined computationally by application of molecular dynamics or finite elements simulations [69,73]. Four key mechanical properties (moduli) of the membrane surrounding a protein result from the transbilayer pressure profile, describing the effect of compression, expansion, bending, and extension [74]. Among these, the area expansion modulus, describing the response of the membrane to in-plane stretching or dilation, and the bending modulus, reflecting the energy with which the curvature of a membrane can be changed, correlate with the tension threshold of MscS gating [75].

The open conformation has a larger membrane cross-section; hence, it is energetically more favorable in a tensed membrane. However, other implementations of the force-from-lipids principle are conceivable for bacterial MS channels. For MscL, the maintenance of an interaction between lipids and an amphipathic helix upon membrane tensing, resulting in an opening movement of the pore helices connected to the amphipathic helix has been proposed [76,77,78], as well as the extent of a hydrophobic mismatch between the membrane-exposed region of the TMD and the surrounding bilayer [79]. For MscS, a change in local membrane curvature has been discussed [35,38,80,81]. Lipids associated with MscS moved to the center of attention with a crystal structure of open MscS that showed aliphatic chains in the pockets at the cytosolic side of the TMD [35]. On this basis, a ‘lipids-move-first’ mechanism has been suggested, proposing that lipids function as ligands that modulate the conformation of MscS by exchanging with the membrane. This paradigm is supported by the insight that the conformational state of MscS in the absence of a membrane can be switched by deliberately removing bound lipids with an excess of detergent. Moreover, the addition of lipids to this delipidated MscS closes the channel again [21]. 

## 4. Coordination of Lipids in MscS and YnaI

### 4.1. Pocket Lipids

In a crystal structure of open MscS [35], aliphatic chains were identified in the pockets between the TM helices, and further studies corroborated the presence of lipids in both the closed and open states. Tryptophan fluorescence quenching provides a reliable method to study lipid accessibility of distinct residues [82,83]. Therefore, in a tryptophan-free MscS mutant, residues lining the pockets were substituted by tryptophan and reconstituted with brominated lipids. The effect of quenching of the introduced tryptophan residues by bromine is an indicator of lipid proximity [84]. A distance below ~9 Å between the tryptophan side chain and the bromine atom can result in quenching [84]. Fluorescence quenching was observed for the residues I150W and F151W in the β domain, the membrane-facing residues L111W, A119W, L123W and F127W on helix TM3b, and the residues A94W, G101W and A103W on helix TM3a [35,39] (Figure 2C). Yet, this approach does not allow for quantitative lipid analysis. Investigation of the complex prepared under the condition that produced the open MscS crystal structure by native mass spectrometry revealed up to five phospholipids bound per complex. Thin-layer chromatography and mass spectrometry identified phosphatidyl ethanol amine (PE) and phosphatidyl glycerol (PG) lipid molecules [35]. An earlier study that utilized size exclusion chromatography and inductively coupled plasma mass spectrometry suggested 2.6–3.1 phospholipid molecules per subunit (corresponding to 18.2–21.7 per complex); here, MscS was supposedly in a closed conformation [85]. The finding that less lipids occupy the pockets in the open conformation agrees with molecular dynamics (MD) simulations [35]. The first cryo-EM structure of MscS reconstituted into nanodiscs showed MscS in a closed conformation, unveiling not only the position of the MscS in the membrane, but also resolving three phospholipids per subunit [19] (Figure 2B,C)–although it is conceivable that more lipids are associated with the complex which are not resolved due to flexibility or motion. Two of the resolved lipids lie parallel to the membrane in the pockets and point towards the pore with their fatty acid tails. The phosphate moiety of one of these lipid molecules is coordinated by R59 through a salt bridge, and its importance is supported by the strong gain-of-function (GOF) phenotype when neutralizing the charge in an R59L mutant: This mutant requires a decreased pressure to open [19,39]. Interestingly, a lipid is still present in this position in a structure of open MscS obtained in the detergent LMNG, indicating that MscS does not need to be wholly delipidated for adopting the open conformation [21]. This conclusion is in agreement with a study by Martinac, Walz, and co-workers [22]: They mimicked membrane tension exerting on MscS embedded in nanodiscs by removing lipids with βCD from the nanodiscs. This way, they propose to obtain an adapted conformation that MscS adopts under sustained tension. Comparison of the closed, open, and adapted state revealed that the volume of the pockets decreases, thus leading to the assumption that even more lipids must be expelled from the pockets in the adapted conformation. 

The structure of YnaI also revealed pockets with lipid-like densities [23,24,25]. Penetration of these pockets by lipids has already been demonstrated earlier by spectroscopic data, as the residues F168W and W201 which are located in the helix TM3b and the β domain, respectively, showed quenching by brominated lipids [86] (Figure 2E). This implies that the two additional TM helices do not shield the pockets from lipids in YnaI and have probably a conserved function like in MscS. However, the presented structures of YnaI do not show the lipid molecule in the same location. In DIBMA-solubilized YnaI, one lipid molecule is found in a similar orientation and position as in MscS. Its phosphate moiety is coordinated by K108 in the cytosolic loop connecting the helices TM1 and TM2, which was backed up by introducing the mutation K108L [23]: This mutant should weaken the interaction with the lipid and showed a GOF phenotype in patch clamp experiments, as less pressure was required for channel opening. Another structure of YnaI, solubilized in the detergent LMNG, showed a lipid molecule closer to the cytoplasmic membrane leaflet that is clamped between the residues R120 and Q100 [24]. Hypoosmotic downshock experiments showed that the mutant R120A resulted in a higher survival rate of *E. coli* compared to the wildtype, supporting the involvement of R120 in lipid binding. This lipid is modelled in a different orientation and position compared to MscS, which is not probed in the earlier spectroscopic experiments [86] (Figure 2E). The latest YnaI structure was obtained with SMA2000, revealing a lipid molecule in the pocket between the helices TM3a and TM2 [25], similar to the one modelled in YnaI solubilized with DIBMA copolymer. A striking difference between these two structures, however, is the position of the lipid’s phosphate moiety, which is in proximity to R202 and W201 in YnaI obtained with SMA2000. No lipid molecule was identified in YnaI solubilized with DDM and reconstituted into Amphipols A8-35 [56]. A thorough comparison of the four YnaI structures revealed that the paddle helix TM2 is tilted by ~15° in YnaI obtained by SMA2000 compared to the other structures [25]. The authors concluded that the position of the phosphate moiety of the lipid is crucial for the geometry of the helix TM2, but it should be noted that also the number of bound lipid molecules would affect the final protein structure. The differently modelled lipid molecules do not rule out each other’s legitimacy but are likely a result of the chosen purification technique. Maintenance of native protein-lipid interactions is therefore desirable, because the number, orientation and position of lipid molecules seem essential for the correct protein conformation.

### 4.2. The ‘Hook’ Lipid

The third lipid molecule observed in cryo-EM structures of closed MscS is clamped between Y27 in helix TM1 and R88 in the periplasmic loop connecting the helices TM2 and TM3, and because of its shape, it was initially termed “hook lipid” [19,20] (Figure 2B,C). It is positioned at the height of the cytoplasmic membrane leaflet but inverted compared to the lipids in this leaflet. The head group is hooked through the periplasmic funnel into the pore. The fatty acid tails of the hook lipid are wedged in between the paddle helices TM1 and TM2, and it was suggested that this impedes the sliding of these helices against each other, which is required for channel opening [22]. One model suggested that this lipid is the initiator of the gating transition and was therefore named “gatekeeper lipid” [22]. According to another model, the hook lipid equilibrates with lipids from the periplasmic leaflet upon opening [21]. Both models are discussed in detail in Section 5.

The shared feature of lipid interaction in the pockets gives rise to the expectation that their function is conserved in MscS-like channels. In the YnaI structures, all densities assigned to lipid molecules are less pronounced than in MscS, either because they are more flexible or because the TMD is weaker resolved than that of MscS. The hook lipid, which is typically best resolved in MscS, however, could not be discovered in any YnaI structure. In YnaI, the inner periplasmic loop connecting the helices TM2 and TM3 is uncharged and shielded from the lipidic environment by the second TM helix bundle of YnaI, and thus it is unlikely to find a hook lipid there. The first periplasmic loop in YnaI is connecting the helices TM-N1 and TM1. This loop, harboring two positively charged residues, is the more likely candidate for the interaction with a potential hook lipid, although, in contrast to MscS, these residues reside in the height of the phosphate moieties of the periplasmic lipids, and not in the height of the cytoplasmic leaflet. Nevertheless, no density in this region could be attributed to a lipid in all YnaI maps so far. As suggested by Catalano et al. [25], this might be because no purification technique applied to YnaI retained all essential lipid molecules, but it is also possible that among the MscS family the hook lipid is a feature unique to MscS. As such, it could play a role for adaptation-which is only observed for MscS so far–as suggested previously [87,88]. 

### 4.3. Pore Lipids

Rod-shaped densities within the permeation pathway were observed in several cryo-EM structures of MscS [19,20,21,22] (Figure 2B,C). In the closed conformation they are only found in the periplasmic side of the pore while in the open conformation also from the cytosolic side densities are seen. They were interpreted as lipid molecules or in replacement as detergent molecules in the solubilized state. Their presence agrees with spectroscopic data: A94 and G101, which line the pore in the periplasmic half of the channel, showed strong quenching by brominated lipids [39]. In the closed state of MscS, L105W, pointing towards the pore axis on the cytosolic side of the pore, and G113W–in the kink separating the helices TM3a and TM3b–were only weakly quenched by brominated lipids [35,89]. The analogous residue in YnaI, L154W, on the other hand was highly quenched, but no pore lipid densities were present in the YnaI structures. The observed quenching effect is of course dependent on the position of the bromine atoms on the lipid tails [39]. It is also possible that the pore lipids in YnaI were harshly removed during purification, or that the observed quenching of L154W is caused by a pocket lipid. 

MD simulations indicated that the pore lipids are necessary for stably closing the pore in MscS completely for ions [20,22]. Additionally, it was shown in MD simulations that the pore lipids move to the periphery of the pore upon opening, and based on the presented cryo-EM map of an adapted state, exhibiting no pore lipids anymore, that the lipids finally leave the pore after opening [22]. Lipids in the pore were also suggested to be involved in the process of adaptation: In the presented scenario, lipids penetrate the gaps that arise between adjacent TM3a helices upon opening, resulting in occlusion of the permeation pathway [88]. This is somewhat supported by an open structure that shows lipid tails extending to these gaps (Flegler et al., 2021). 

## 5. State-of-the-Art Models for Lipid-Driven Gating of MscS

The mechanism of how membrane tension leads to MscS channel opening remained elusive although crystal structures showing MscS in a closed and an open conformation were acquired [29,30,36,40] and several computational approaches were applied to understand the conformations seen in the crystal structures [90,91,92,93]. The lipids-move-first model emerged with the identification of acyl chains in the pockets of a crystal structure, which were the basis for further experiments encompassing MD simulations, native mass spectrometry, and tryptophan fluorescence quenching [35]. These experiments indicated that the pockets of MscS are indeed accessible to lipids and that upon opening, lipids are likely to be expelled from the pockets as the pocket volume decreases. Further quenching experiments suggested that the pocket lipids lie perpendicular to the bilayer lipids, as residues on helix TM3b, but also on the adjacent β-domain, were quenched by brominated lipids [39]. The associated lipids in MscS became visible in cryo-EM structures of MscS and based on their presence or absence in certain conformations, molecular mechanisms for MscS gating were proposed [19,20,21,22]. 

### 5.1. The Hook Lipid as Gating Initiator

The initiation of gating in the model presented by the groups of Martinac and Walz is based on the absence and presence of the hook lipid, which was therefore termed “gatekeeper” lipid [22] (Figure 5B). Coordinated by R88 in the TM2-TM3 loop and Y27 in TM1, the tails of the lipid are wedged between the helices TM2 and TM1. This arrangement was suggested to impede sliding of adjacent TM1-TM2 bundles against each other, which is a requirement for the opening transition. The experiments by Zhang et al. [22], using βCD to imitate membrane tension within MscS-containing nanodiscs, supported the idea that the hook lipid is the first lipid to leave the channel, because a short incubation time with βCD showed MscS still in a closed conformation, but with this lipid missing. It was proposed to initiate the gating transition, and therefore named “gatekeeper lipid”. Incubation with βCD for 16 h resulted in an adapted state (denoted as desensitized by Zhang et al.), as the tension, caused by lipid removal, persisted. Elucidating the molecular basis underlying an adapted state of MscS marked a key finding, as this behavior was known for a long time [41,42] and has been studied extensively [34,91,94,95,96,97], but there is disagreement on its structural cause [43,88,93]. Zhang et al. observed the open conformation by using the MscS mutant G113A, a residue located in the pore helix TM3a, which was shown to abolish adaptation [22,97]. The mutations of R88 to a tryptophan or a serin residue resulted in loss-of-function (LOF) channels that are more difficult to open in patch clamp experiments and have decreased survival rates in hypoosmotic downshock experiments [89,97]. This does not support the assumption that the hook lipid is strictly required for gating initiation, because without this lipid, or with a weaker interaction with this lipid, MscS should be easier to activate.

The obtained states–closed, open, and adapted–show a decreasing pocket volume, leading to the interpretation that the removal of an increasing amount of pocket lipids following the hook lipid dissociation drives MscS successively into the different conformations [22]. This gating model was probed by MD simulations showing that the pore lipids are required to completely close MscS for ions and move to the periphery of the pore upon channel opening. This is in agreement with detergent densities observed in an open structure of MscS in LMNG, which were suggested to take the place of lipid molecules [21]. As no pore lipids were observed in the map of the adapted state, the model proposes that the pore lipids ultimately leave the pore upon persisting tension [22].

### 5.2. Gating Initiation by Pocket Lipid Removal

The hook lipid is located in the height of the cytoplasmic region but inverse to the lipids of the cytoplasmic membrane leaflet. The region of highest tension according to the transbilayer pressure profile is at the beginning of the fatty acid chains, and here also the changes in the pressure profile are highest upon tension [68,98,99]. The hook lipid contacts this particular region only with the end of its fatty acid tails, which renders it unlikely to initiate the gating transition. Furthermore, it is for energetic reasons unplausible how the hydrophilic phosphate moiety of the hook lipid could dissociate into the hydrophobic core of the cytosolic leaflet as presented in the model by Zhang et al. [22]. Thus, a model was proposed based on the ligands observed in the closed and open structures of MscS [21] (Figure 5C). Because the structure of open MscS was obtained in LMNG, it must be considered carefully which detergent molecules could be exchanged with lipids in a membranous environment. It is conceivable that the lipids in the grooves and pockets of MscS initiate the gating because they show a weaker resolution than the hook lipid, underpinning that the lipids in the region where the cytoplasmic leaflet is attached are the most dynamic ones and should therefore be the easiest to be removed by membrane tension. This is concomitant with the transbilayer pressure profile because the tension is increased in this region [68,99]. Here, the opening of MscS is a consequence of the lipid extrusion from the grooves and pockets into the cytosolic leaflet to compensate for the decrease in the volume of the pockets. While the diameter of the TMD does not change on the cytosolic side, it increases significantly on the periplasmic side. This generates gaps between adjacent sensor paddles, and lipids from the periplasmic leaflet are likely to enter these gaps to shield the bulk water in the channel from the surrounding lipidic environment. These lipids can then equilibrate with the hook lipid, which is in the height of the periplasmic membrane leaflet surrounding MscS in its open conformation.

In an alternative hypothesis, the hook lipid launches the gating transition when it is positioned at the height of the cytosolic leaflet due to the changes in the membrane tension: This might pull the hydrophobic tails and loosen the interaction between adjacent sensor paddles that are mediated by the phosphate moiety of the hook lipid. When the interaction is lost the paddles tilt and enable the equilibration of the hook lipid with the periplasmic leaflet along the newly formed inter-paddle grooves. The rearrangement of the paddles moves the cytosolic pockets towards the membrane and the pocket lipids equilibrate with the cytosolic leaflet powered by the changing tension profile. Removal of the pocket lipids enables the outwards movement of the pore helix TM3a and completes gating.

## 6. Modulation of Membrane Properties

### 6.1. Bilayer Thickness

The biophysical properties of the membrane bilayer influence the gating properties of the embedded MS channels [81]. This is directly linked to the transbilayer pressure profile of the lipid bilayer and the force-from-lipids principle stating that deformation of the pressure profile results in changes in the channel gating properties [68,71]. Hence, as different lipids have unique characteristics, the lipid composition of the membrane will directly affect channel gating [100,101]. Such lipid characteristics include the head group properties, acyl chain lengths, and degree of acyl chain saturation of phospholipids, but certainly also contributions of other lipids like cholesterol. Already in 2005 Moe and Blount showed that the replacement of PC by PE results in a higher activation threshold of MscL [66]. 

Reduction of bilayer thickness–as it occurs under membrane tension–and thereby increasing the extent of protein-bilayer hydrophobic mismatch was shown to lower the activation threshold for MscL [59], but MscS is less sensitive to bilayer thinning [62]. Indeed, a cryo-EM structure of MscS reconstituted into nanodiscs comprising didecanoyl phosphatidylcholine (PC-10), creating an unphysiologically thin bilayer of 15.8 Å [102], showed only a subconducting state, while MscS in dimyristoyl phosphatidylcholine (PC-14) and dilauroyl phosphatidylcholine (PC-12) still resulted in closed MscS [22]. These experiments show that hydrophobic mismatch alone is not sufficient to open bacterial MS channels. 

### 6.2. Polyunsaturated Fatty Acids

Bacteria adapt their membrane lipid composition to changes in their surroundings. As an example, the amount of unsaturated fatty acids increases with decreasing temperature, while an increase in temperature results in more saturated fatty acids, because the temperature greatly affects the fluidity of the membrane [103]. Polyunsaturated fatty acids (PUFAs) can pack less tightly than saturated or monounsaturated fatty acids (MUFAs) and thus, lead to a more fluid membrane [104]. Martinac and coworkers reconstituted MscS and MscL into artificial bilayers composed of MUFAs or PUFAs to investigate the effect of these lipids on the gating properties of the channels, and to experimentally assess the transbilayer pressure profile using a combination of patch clamp experiments with nuclear magnetic resonance [70]. While in pure polyunsaturated lipid bilayers MscL is more sensitive during opening and closing, the activation of MscS is not affected, but lower pressures are required for closing. This was explained by an increasing degree of unsaturation resulting in bilayer thinning, which lowers the activation threshold of MscL. In addition, PUFAs do not affect the opening process of MscS but stabilize its open conformation. The effect of PUFAs on MscS further shows that changes in the transbilayer pressure profile of the less rigid unsaturated bilayer are the main factor for MscS gating: Because of the disproportionate changes in diameter at the cytosolic and periplasmic side of the TMD upon opening, the closed conformation fits the transbilayer pressure profile of an unsaturated bilayer less well than the open conformation. Brominated lipids used for tryptophan fluorescence quenching have similar characteristics as monounsaturated lipids, as the bulky bromine atom has a similar effect as a *cis* double bond [82]. This is an important note as it suggests that brominated lipids do not alter the gating behavior of MscS observed in spectroscopic experiments.

### 6.3. Cardiolipin

Another special lipid is the anionic cardiolipin (CL), which is unique among phospholipids as it has four fatty acid chains. Typically, CL makes up 2–8 % of the total lipids in *E. coli* [105], but the proportion of anionic phospholipids, including CL, is increased when bacteria are grown in media of higher salinity [106]. The acyl chain composition of CL is also affected by the salinity [107], and cells lacking the enzyme CL synthase show increasing growth defects upon elevated salt concentrations of the growth medium [108], implying involvement of CL in bacterial osmoregulation. CL is concentrated at the poles of the cells [106] but the accumulation of MscS at the cell poles is controversial [109,110]. Spectroscopic studies indicated that CL is essential for proper MscL gating and directly interacts with the channel [111,112,113], which, among other factors, might explain the role of CL for osmoregulation. The effect of CL on MscS mechanosensitivity was investigated in patch clamp experiments by enriching MscS-containing proteoliposomes with CL [114]. The application of CL increased the gating frequency of MscS, and the authors reasoned this is caused by a characteristic lateral pressure profile of CL: CL induces negative curvature and CL-containing membranes are stiffer and more ordered, which stabilizes the closed conformation of MscS. This agrees with the finding that in a defined PE/PC lipid environment, the presence of CL increases the activation threshold of MscS. Although it was shown that CL molecules can penetrate the hydrophobic pockets of MscS [39], native mass spectrometry could not identify adducts with a mass range fitting CL lipid molecules [35]. Hence, the effect of CL on MscS is likely contributed to changes in the membrane properties than to direct protein-lipid interactions. Noteworthy, one study on YnaI identified an enrichment of CL and phosphatidylserine (PS) in a lipid extract derived from purified YnaI using thin layer chromatography [24]. The authors hypothesized that upon tension, CL is pulled out of the pockets in YnaI, equilibrates with the membrane, and, because of its negative curvature-inducing property, alters the local membrane curvature around YnaI in favor of channel opening. However, there is no structural evidence for bound CL molecules in the pockets of YnaI.

### 6.4. Cholesterol and Membrane Stiffness

Cholesterol is not native to bacterial membranes but was employed to alter membrane properties in MS channel studies. It was shown that cholesterol influences the physicochemical properties of a bilayer, as well as the functionality of ion channels [115,116]. In particular, it was shown that cholesterol affects the bilayer thickness [117,118,119], and substantially increases membrane stiffness (also referred to as elastic area expansion modulus) due to higher acyl chain order [120,121]. Additionally, incorporation of cholesterol results in the alteration of the transbilayer pressure profile, thereby shaping membrane protein function as well [99]. Patch clamp recordings on MscS and MscL reconstituted into Azolectin liposomes showed that upon doping the membrane with cholesterol, the activation threshold for both channels increases, but this effect is more pronounced for MscS [62]. For MscL the observed effect can be explained by its sensitivity towards bilayer thickness [59]. In the case of MscS, the highly increased activation threshold was explained by the higher degree of membrane stiffness [62]. The Martinac group further investigated the effect of stiffer membranes on MscS using branched or fully saturated lipids [75]. The authors systematically doped Azolectin liposomes with stiffer lipids: Addition of PE(18:1), known to be a stiffer lipid than PC(18:1) [122], resulted in a higher pressure threshold required for channel opening [75]. Adding a branched or a fully saturated PE form to Azolectin liposomes increased the activation threshold even more. This trend was also observed for analogous experiments with PC. By comparing the effects of polyunsaturated and monounsaturated PC–which differ in their bending modulus but not in their area expansion modulus–the authors concluded that the key factor for the tension sensitivity of MscS is the area expansion modulus, the stiffness of the membrane, and not the bending modulus. Furthermore, tryptophan fluorescence quenching experiments revealed that cholesterol, unlike CL, cannot penetrate the pockets of MscS [39].

### 6.5. Lysophosphatidylcholine and Other Amphipathic Compounds

Amphipathic molecules, like lysophosphatdylcholine (LPC) are known to facilitate MS channel gating when intercalating asymmetrically in only one membrane leaflet [59,60,61,63,67]. The finding that amphipaths can activate MS channel is in accordance with the force-from-lipids principle, as membrane deformation results in structural rearrangements of an embedded channel [67]. Like with cholesterol, the effect on MscS is bigger than on MscL, as the activation threshold of MscS is more decreased upon addition of LPC than that of MscL [62]. Because it has only one fatty acid tail, LPC is cone shaped and induces bending of the membrane when added to one side. Its activation ability was attributed to this ability to induce membrane curvature, because it would mimic the curvature of membrane patches during patch clamping [59,80]. In patch clamp experiments, MscS and MscL could also be activated when positive pressure was applied, and with decreasing curvature upon positive pressure, the channel activity increased. Hence, curvature as such is probably not the driving force for gating, but it is likely that LPC rather directly creates membrane tension and the associated deformation of the transbilayer pressure profile drives the structural reorganizations [62,123,124]. This is supported by the finding that LPC could activate MscS no matter in which of the two leaflets it was incorporated [35]. The lipid-accessible pockets of MscS provide an alternative mechanism for LPC-mediated gating: Because the pockets are also penetrated by LPC molecules, it was suggested that LPC can migrate from the cytosolic leaflet into the pockets, and the lesser hydrophobic volume of LPC destabilizes the closed conformation [35]. 

Experiments tackling the gating mechanisms of MscS and MscL have shown that both can be fully activated by the application of LPC [59,60,62,125,126,127,128,129], and that the single-channel properties of MscS activated by LPC did not differ from tension-activated characteristics [90]. On the other hand, another study found LPC-activated MscS in a sub-conducting state [35], and it was also not possible to obtain open YnaI by a combination of LPC and DIBMA copolymer [23].

Antimicrobial peptides are typically positively charged amphipathic molecules which have gained attention as potent alternatives to classic antibiotic compounds [130,131]. Among these, (ethyl and butyl) piscidins were shown to modulate the gating of MscS [16,132]. Piscidins were suggested to cause local lipid redistribution, which would also alter the membrane properties within the microenvironment of the MS channel and thus, change its gating behavior [133]. Other amphipathic compounds, like (propyl and ethyl) parabens were shown to lead to changes in the lateral pressure that occur upon insertion of the molecules into the bilayer [134,135]. 

Membrane microdomains, that differ in their local lipid composition–so-called lipid rafts–were discovered in bacteria several years ago [136,137]. They are typically enriched in certain lipid species, e.g., PE, CL, or lysolipids, but can also comprise more rare lipids like the sterole-like hopanoids or terpenoids like staphyloxanthin [138,139]. We did not find studies that specifically investigate the role of bacterial lipid rafts on MS channels, but it might be reasonably assumed that the gating behavior of MscS homologues within lipid rafts shows differences, because each lipid species comes with certain properties that affect the lateral pressure profile of the membrane, which modulates the gating of MscS.

## 7. Conclusions and Outlook

Lipids influence the function of MS channel in two different ways: As ligands, they interact with a specific region, which modulates the gating of the channel. Additionally, certain lipid components alter membrane properties in favor or in disfavor of channel activation. The bacterial examples of MS channels mentioned in this review, MscS and MscL, demonstrate that not all MS channels need to follow a universal mechanism but that sensitivity towards membrane tension can result from different lipid-protein interactions and membrane properties. Typically, different experimental approaches suggest more than one tension-sensing model, especially for the MscS-like channels. Often these models do not exclude each other, and a cumulative effect seems likely. It will be interesting to see how future experiments can dissect contributions from different lipid-protein interactions to the overall tension sensing of MscS-like channels.

## Figures and Tables

**Figure 1 ijms-23-15071-f001:**
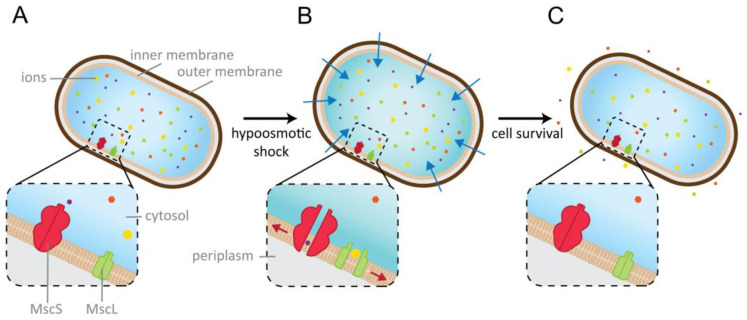
Physiological role of mechanosensitive channels in prokaryotes. (**A**) The bacterial MS channels MscS and MscL are located in the inner membrane and play an essential role for the osmoregulation of the cell. (**B**) A rapid drop of the osmolarity of the environment leads to water influx (blue arrows) that increases the turgor and, as a result, the membrane tension (red arrows). MscS and MscL sense these elevated membrane tension levels and open transiently to release osmolytically active solutes. (**C**) When the osmotic balance is restored, the channels close again.

**Figure 2 ijms-23-15071-f002:**
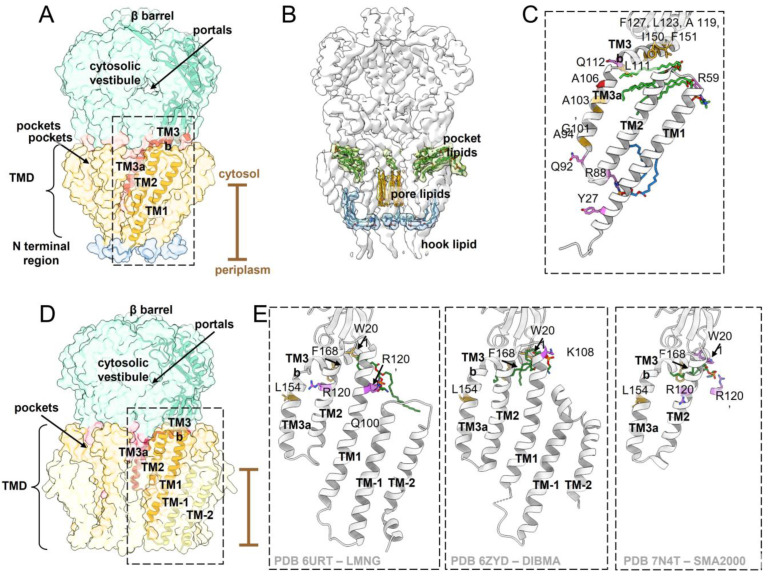
Architecture and lipid coordination of MscS and YnaI. (**A**) The surface of MscS in its closed conformation (PDB 6RLD [19]) is shown in transparent colors, and one subunit is additionally depicted as ribbon. The cytosolic vestibule is colored in green, the kinked pore helix TM3 in red, the sensor paddle helices TM1 and TM2 in orange, and the N terminal region in blue. The helices TM1, TM2, and TM3a comprise the transmembrane domain (TMD). The position of the membrane is indicated by the brown bar. (**B**) The EM density for MscS (EMDB 4919 [19]) is shown in white, and the densities of bound lipid molecules are shown in transparent colors with the stick model. The pocket lipids are depicted in green, the pore lipids in orange, and the hook lipids in blue. For better visibility, a gaussian filter was applied to the map. (**C**) The section highlighted in the dashed box in (**A**) is shown and specific residues are shown as sticks. The model is shown as white ribbon; pocket lipid molecules are shown in green, and the hook lipid in blue. Pocket residues in contact with lipids are colored yellow, residues that are probably involved in lipid binding are purple, and residue A106 is red. (**D**) For the larger paralog YnaI (PDB 6URT [24]) the same depiction and color code was used as for the MscS structures. YnaI has a second set of sensor paddles (yellow), that is shifted against the inner helix bundle (TM-1 (“minus 1”) and TM-2 (“minus 2”); (**E**) The section in the dashed box indicated in (**D**) is shown for three independently obtained YnaI models (left: PDB 6URT [24], middle: PDB 6ZYD [23], right: 7N4T [25]) and residues are colored as described for MscS in (**C**). Only the residues interacting with the lipid as described in the corresponding studies are depicted in purple. Pocket lipid molecules (green) for YnaI were observed in different positions and orientations in YnaI.

**Figure 3 ijms-23-15071-f003:**
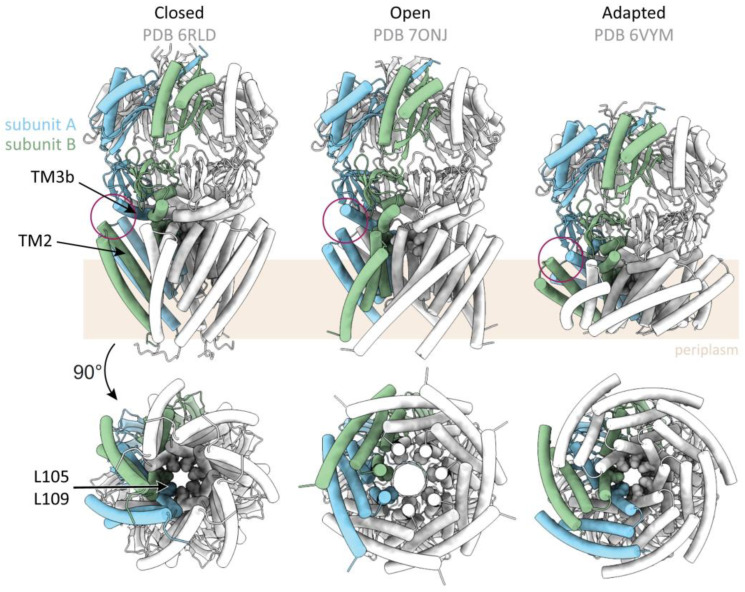
Structures of distinct functional states of MscS. Three different conformational states of MscS were observed so far on a structural level: the closed state (left; PDB 6RLD [19]), a conducting state (middle; PDB 7ONJ [21]), and an adapted one (right; PDB 6VYM [22]). The three structures are shown in the same orientation, with one subunit depicted in green and one in blue, either viewed from the side (**top**) or along the pore axis from the periplasmic side (**bottom**) The membrane is indicated by the pale brown rectangle. The hydrophobic sealing ring of the pore, which is comprised of the two Leucine residues L105 and L109, is shown in ball representation. Upon opening, the sensor paddles increase their tilt in the plane of the membrane and the pore helix TM3a shifts outwards (**middle**). Additionally, the latter rotates around its own helical axis, which withdraws the locking leucine residues from the pore axis, resulting in a conducting state. The sensor paddles of the presented adapted state (**right**) are tilted even more within the membrane plan, which leads to a flattened TMD, while the pore is occluded by the sealing residues. Concluding from the existing structures, it was proposed that the state of the pore is related to the alignment of the helices TM2 and TM3b (pink circle): If there is an interaction between helices from different subunits, the channel is non-conducting, as observed in the closed and adapted conformation. If the interaction takes place between the helices TM2 and TM3b of the same subunit, the pore is in a conducting state.

**Figure 4 ijms-23-15071-f004:**
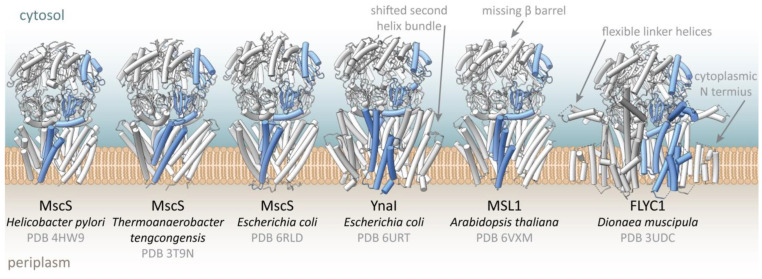
Structures of MscS homologues. The structural organisations of different MscS-like channels are shown in light grey, highlighting both similarities and differences between the channels. The structures are shown in the same size and orientation, and the same subunit of each channel is emphasised in blue. Besides, their approximate position in the membrane (brown), as suggested from cryo-EM structures, is considered in this figure. Structures of MscS were so far obtained from three different organisms, all exhibiting a highly similar architecture. Additionally, several structures of the *E. coli* paralogue YnaI were obtained, as well as structures from the plant channels MSL1 and FLYC1. The most obvious peculiarities of the larger homologues are indicated by grey arrows. Details and references for the depicted models are listed in Table 1.

**Figure 5 ijms-23-15071-f005:**
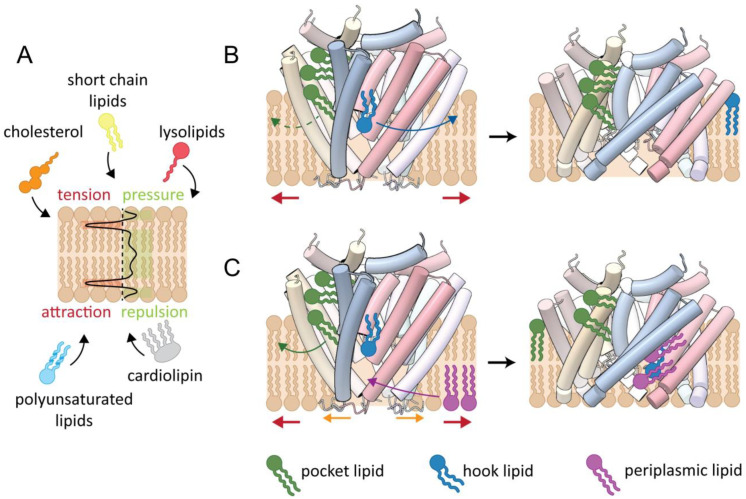
Lipids-move-first gating models for MscS as an implementation of the force-from-lipid principle. (**A**) The transbilayer pressure profile is shown for an ideal lipid bilayer. There is repulsion between the phosphate moieties and the fatty acid moieties, and attraction between the lipids at the onset of the fatty acid chains. This pressure profile is affected by embedded proteins and by the composition of the membrane, which allows a targeted modification of the transbilayer pressure profile. (**B**,**C**) For the illustration of two different gating models, MscS is shown in its closed (left, PDB 6RLD [19]) and open (right, PDB 2VV5 [40]) conformation with individually colored subunits. In the two models, increasing membrane tension (red arrows) leads to different immediate reactions involving certain lipids. For clarity, only one set of interacting lipid molecules is depicted. (**B**) Membrane tension leads to the dissociation of the hook lipid (blue arrow). This enables adjacent TM1-TM2 helix bundles to slide against each other, which is a prerequisite for channel opening. The hook lipid flips and enters the cytosolic leaflet. In this model, extrusion of pocket lipids is predominantly relevant for adopting subconducting- and adapted states (dashed green arrow). (**C**) Gating is not initiated by leaving of the hook lipid, but by lipids bound in the gaps and pockets at the cytosolic side (green arrow). Lipids from the periplasmic leaflet enter the gaps at the periplasmic side which appear upon channel opening, as the N terminal domains move outwards (orange arrows) and equilibrate with the hook lipid. This way, they prevent the exposure of the bulk water in the funnel to the lipidic surrounding.

## Data Availability

Not applicable—The data can be found in the cited literature.

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
