# Peer review of "How Functional Lipids Affect the Structure and Gating of Mechanosensitive MscS-like Channels"

_ijms, 2022, doi:10.3390/ijms232315071_

Round 1

Reviewer 1 Report

This review is well written and explains the complex states and membrane interactions between MscS channels. The paper is valuable and insightful.

I suggest consolidating the citations such that they all are in the same numerical format (L 243, 399, 427, 449, 455, 459, 484)

Figure 1 shows blue arrows not as written, red arrows

Fig. 2 A, B and D show poor graphical renditions. The PDB website offers much better structure renditions and it is highly recommended to adjust resolution and contrast.

I am not convinced that the expression ‘purple sticks’  (L109 and elsewhere) is the best way to describe lipids.

The term nanodisk should be explained as there are different versions available.

L158: replace ‘much more’ by a numerical/quantitative expression as in L185

Fig. 4: Instead of ‘Structures of MscS and -homologues’ use ‘Structures of MscS-homologues’

L246 ff: unclear why text is in bold face.

L258: by tensing the membrane – better by stretching the membrane

Caption of Fig. 5 is confusing. The grouping of A and B  and then B separately is not delivering a clear description.

I suggest replacing different, poorly discernible colors by dashed or dotted arrow lines.

L286: the term headgroups is ambiguous – use polar or phosphate moiety.

L324: This sentence should be rewritten: but also resolving three phospholipids per subunit.

The fact that ‘lipid rafts’ determine membrane domains provides another take on the control of MscS channels – namely that the lipid composition determines the force threshold required to switch to the open or adapted configuration. Perhaps this concept can be included in the discussion.

Reviewer 2 Report

See attached report.pdf file
